
# Factors controlling atmospheric DMS and its oxidation products (MSA and nssSO$_4^{2-}$) in the aerosol at Terra Nova Bay, Antarctica

Silvia Becagli[1,2], Elena Barbaro[2], Simone Bonamano[3], Laura Caiazzo[1], Alcide di Sarra[4], Matteo Feltracco[2], Paolo Grigioni[4], Jost Heintzenberg[5], Luigi Lazzara[6], Michel Legrand[7], Alice Madonia[3], Marco Marcelli[3], Chiara Melillo[6], Daniela Meloni[4], Caterina Nuccio[6], Giandomenico Pace[4], Ki-Tae Park[8], Suzanne Preunkert[7], Mirko Severi[1,2], Marco Vecchiato[2], Roberta Zangrando[2] and Rita Traversi[1,2]

[1]Department of Chemistry "Ugo Schiff", University of Florence, Sesto F.no (FI), 50019, Italy.
[2]Intitute of Polar Sciences, National Research Council (CNR-ISP), Venice Mestre (VE), 30172, Italy
[3]Department of Ecologic and Biologic Science,University of Tuscia, Civitavecchia, 00053, Italy.
[4]ENEA Laboratory of Observations And Measurements for the environment and climate, Rome, Italy
[5]Leibniz-Institute for Tropospheric Research, Leipzig, 04318, Germany.
[6]Department of Biology, University of Florence, 50121, Florence, Italy
[7]IGE-Institute des Géosciences de l'Environment, S. Martin d'Heres, 38400, France
[8]Korea Polar Research Institute (KOPRI), Incheon 406-840, South Korea.

*Correspondence to*: Silvia Becagli (silvia.becagli@unifi.it)

**Abstract.** This paper presents the results on simultaneous high time resolution measurements of biogenic aerosol (methane sulfonic acid-MSA, non-sea salt sulfate nssSO$_4^{2-}$) with its gaseous precursor dimethylsulfide (DMS) performed at the Italian coastal base Mario Zucchelli Station (MZS) in Terra Nova Bay (MZS) during two summer campaigns (2018-2019 and 2019-2020). The study provides information on marine biological activity in the nearby polynya in the Ross Sea and on the influence of biogenic and atmospheric processes on biogenic aerosol formation. Data on atmospheric DMS concentration are scarce especially in Antarctica. The DMS-maximum at MZS occurs in December, one month earlier than at other Antarctic stations. The maximum of DMS concentration is connected with the phytoplanktonic senescent phase following the bloom of Phaeocystis antarctica that occurs in the polynya when sea ice opens up. The second plankton bloom occurs in January and, despite the high Dimethylsulfopropionate (DMSP) concentration in sea water, atmospheric DMS remains low probably due to its fast biological turnover in sea water in this period. The intensity and timing of the DMS evolution during the two years suggests that only the portion of the polynya close to the sampling site produces a discernible effect on the measured DMS. The closeness to the DMS source area, and the occurrence of air masses containing DMS and freshly formed oxidation products allow to study the kinetic of biogenic aerosol formation and the reliable derivation of the branch ratio between MSA and nssSO$_4^{2-}$ from DMS oxidation that is estimated to be 0.84 ±0.06. Conversely, for aged airmasses with low DMS content, an enrichment of nssSO$_4^{2-}$ with respect to MSA, due to the presence of background concentration of nssSO42- from volcanic origin (Erebus) or from long range transport, takes place. Therefore, the aged air mass presents an MSA/ nssSO$_4^{2-}$ ratio lower than in newly formed biogenic aerosol. By considering the sum of MSA and biogenic nssSO$_4^{2-}$, we estimate that the mean contribution of biogenic particulate matter to PM$_{10}$ is 17%, with a maximum of 56%. The high contribution of biogenic aerosol to the total PM$_{10}$ mass in summer in this area highlights the dominant role of the polynya on





biogenic aerosol formation. Finally, due to the regional and year-to year variability of DMS and related biogenic aerosol formation, we stress the need of long-term measurements of atmospheric DMS and biogenic aerosol along the Antarctic coast and in the Southern Ocean.

## 1 Introduction

Dimethylsulfide (DMS) is a volatile organic sulfur compound produced in surface oceanic waters all over the world
characterized by low solubility in water. DMS is formed in the breakdown of the dimethylsulfoniopropionate (DMSP), a phytoplankton metabolite (Stefels, 2000). Approximately 10% of total global DMS production ventilates through the sea-air interface (Simó et al., 1999; Simó and Pedrós-Alió, 1999) to the atmosphere, where it accounts for approximately 50% of the natural global sulfate burden (Simó, 2001). The global DMS flux to the atmosphere is currently estimated to be 28.1 (17.6–34.4) Tg S per year, which is approximately half the anthropogenic global atmospheric sulfur input (Klimont et al., 2013;
Lana et al., 2011). This makes DMS an important contributor to global sulfur fluxes. Once in the atmosphere, DMS is oxidized by the hydroxyl (OH), nitrate ($NO_3$), and bromine oxide (BrO) radicals to form either methanesulfonic acid (MSA) or sulfur dioxide ($SO_2$), which is further oxidized to $H_2SO_4$ (Gondwe et al., 2003; Read et al., 2008). The production of sulfuric acid and MSA (having low vapor pressure) may lead to new particle formation (NPF) when few particle condensation nuclei are available (Dall'Osto et al., 2017). NPF linked to DMS products may play a fundamental role in the
polar regions, with possible effects on climate (Dall'Osto et al., 2017). The growth of particles following NPF is crucial in generating cloud condensation nuclei (CCN), which eventually allow the formation of cloud drops.  As CCN are important for cloud formation and thereby indirectly affect the radiation balance, they have an important climatic impact and are involved in feedback processes (Charlson et al., 1987). Actually, there are still large uncertainties in both the sign and the amplitude of this feedback (Quinn and Bates, 2011). Besides, model calculations of the future response of DMS to changes
in global temperature vary widely: both increases (Cameron-Smith et al., 2011; Gabric et al., 2005; Qu et al., 2021; Wingenter et al., 2007), and decreases (Kloster, 2007)in surface water DMS concentrations have been predicted.

DMS concentrations global climatology shows that the polar regions are of significant importance to the total global DMS production, in particular the Southern Ocean (Gondwe et al., 2003; Lana et al., 2011). The total annual Southern Ocean (south of 40°S) DMS flux is estimated at approximately 5.8 Tg S (Kettle et al., 1999; Lana et al., 2011). Most of the DMS
emission, 3.4 Tg S (Jarníková et al., 2016), occurs during summer months (December to February).

The link between climate change and DMS production is complex and involves a great number of oceanic and atmospheric processes: in polar region the maximum DMS concentrations in the water occurs in early summer and is primarily associated with sea-ice break-up (Stefels et al., 2018). Although the retreat in sea ice will directly impact on the release of DMSP and DMSP-producing algae, changes in the physical environment can also indirectly impact on phytoplankton productivity and
composition through changes in light and nutrient availability (Ducklow et al., 2007; Montes-Hugo et al., 2009).





Wind speed plays a relevant role in DMS production in the ocean and in regulating the flux from the ocean. The depth of the oceanic mixed layer, largely influenced by wind, is crucial for the determination of oceanic DMS distribution: for example, a high wintertime ice extent can shield the water column from high wind speeds, thus preventing the deepening of the winter mixed layer. Declining wind speeds in summer can cause the persistence of a shallower mixed layer depths, and when these
variables coincide with the seasonal summertime increase in light availability for primary production, high DMS summer maxima are observed (Saba et al., 2014). Similarly, high summertime winds or a shorter duration of the sea ice season along the marginal ice zone can lead to lower summertime chlorophyll-a (Chl-a) maximum as the mixed layer is deeper, thereby inhibiting algal cells from overcoming light limitation.

Besides processes in the water column, winds affect ocean-air DMS fluxes in a nonlinear way. As wind speed increases
until 10 m/sec DMS transfer rates increase, then it decreases due to the amphiphilic nature of DMS that leads to transfer delay because higher wind speeds cause a higher concentration of sinking bubbles by whitecapping of the ocean surface (Vlahos and Monahan, 2009).

Several studies tried to correlate DMS concentrations in sea water and algal biomass from surface and satellite data, but contradictory results have been found : positive (Andreae and Barnard, 1984; Belviso et al., 2004; Leck et al., 1990),
negative (Froelich et al., 1985; Watanabe et al., 1995), as well as absent (Barnard et al., 1984; Deng et al., 2021) correlations were found. Such uncertain relationships affect the calculation of DMS flux based on the algal biomass. In most models, the DMS fluxes are obtained from small or medium scale field observations (Gabric et al., 2014; Kloster et al., 2006) making our understanding of the mechanisms controlling DMS emissions regionally dependent; however, some regions of the Southern Ocean are not covered by measurements, making these estimates unpractical.

Especially, data on DMS concentrations in the Ross Sea are scarce, and atmospheric DMS observations are missing. Actually, the study of DMS production and its fate in the atmosphere is relevant in this area due to the presence of a persistent polynya. Polynyas are areas of seasonally open water surrounded by sea ice in high latitude regions. They are characterized by abundance of macronutrients, an ample supply of iron (Fe) from melting sea ice and/or glaciers and continental shelf sediments resulting from the intrusion of relatively warm, salty, and nutrient-rich Circumpolar Deep Water
(Arrigo et al., 2012; Sherrell et al., 2015; St-Laurent et al., 2017). Consequently, they often exhibit high primary productivity (Arrigo et al., 2012; Arrigo and van Dijken, 2003; Yager et al., 2012) because they are the first polar marine systems to be exposed to the increasing springtime solar radiation (Arrigo and van Dijken, 2003; Criscitiello et al., 2013). The polynyas in the Southern Ocean are the most productive biological regions and have the highest DMS sea–air flux in the world (Kettle et al., 1999). DMS concentrations as high as several hundred nanomoles per liter have been observed in polynyas along the
coastal regions of Antarctica, such as the Ross Sea and Amundsen Sea (Tortell et al., 2012; del Valle et al., 2009) where there are the favorable conditions for Phaeocystis antarctica blooms (Oliver et al., 2020).

Finally, the large uncertainty in the processes surrounding DMS production emphasizes the need for an improved mechanistic understanding and model parameterization of the atmospheric DMS. However, measurements of DMS in the atmosphere are scarce and especially in coastal Antarctica because of the difficulty in conducting a field observation in these





extreme environments. Therefore, the sources and the evolution of the aerosol over the Antarctic are still a subject of many

open questions. It is necessary to fill the data gap in the knowledge of biogenically-derived aerosols in the Antarctic to

improve understanding of the effects of ocean ecosystem on the marine aerosol-cloud-climate system.

In this study we report high resolution (12h) measurements of MSA and $nssSO_4^{2-}$ in the particle phase simultaneous to gas

phase DMS obtained for the first time in Northern Victoria Land, at the Mario Zucchelli Station (MZS) facing the polynya

area in the Ross Sea.

This work gives new hints to enhance our knowledge of the interactions between oceanographic parameters, the surface

ocean biosphere, and biogenic aerosol formation in this region of our planet.

## 2. Methods

### 2.1 Sampling area

Aerosol and DMS sampling are performed in two Antarctic summer campaigns (AC): 2018-2019 and 2019-2020. The

campaigns lasted from the beginning of November until to the end of January in the area surrounding the Antarctic Italian

base Mario Zucchelli Station (MZS – 74°42' S, 164°07' E) located at Terra Nova Bay, along the coast of the Northern

Foothills to north-east of Gherlache Inlet (Fig. 1). There is a persistent polynya in the sea facing the base, the extent of which

is shown in the average ice maps for the two campaigns (Fig. 1). Fig. 1 also includes average DMS concentrations (nM) in

surface waters for the period December-January according to the climatology of (Hulswar et al., 2021).

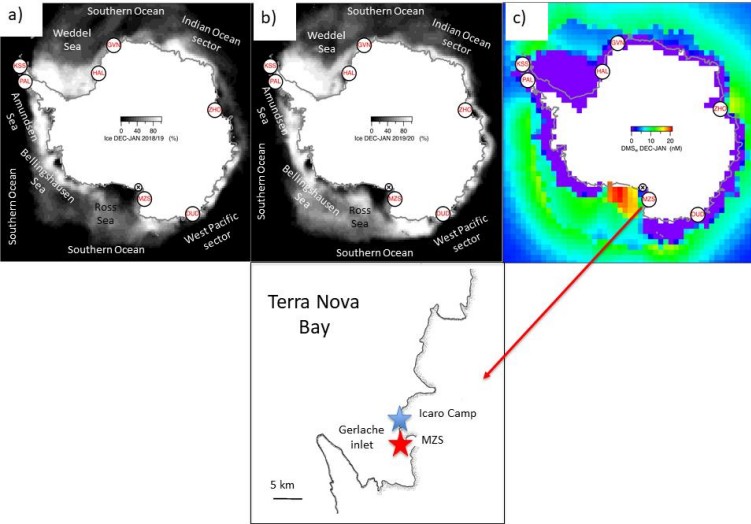

**Figure 1.** Average sea ice cover during campaign 2018/19 (a), and 2019-2020 (b) with sea ice data from NSIDC (https://nsidc.org, last accessed 2021.11.04); c) average DMS (nM) in surface waters during December-January according to the DMS climatology of (Hulswar et al., 2021). Figure on the top reports the enlarged map of the Terra Nova Bay with Mario Zucchelli Station (MZS- red





star) and the aerosol samplings site (Campo Icaro- light blue star). The geographical location of the Antarctic stations reported in
Table 1 are reported in figure (a), (b) and (c).

## 2.2 Aerosol sampling and analysis

In order to avoid possible contamination from the base, the site chosen for aerosol sampling is Icaro Camp (74° 42' 43"S
164° 07' 00"E) located about 2 km south of MZS. The aerosol sampler was installed on the hill facing the sea at about 30m
above sea level.

Aerosol sampling was performed at 12 h resolution by a low volume sequential aerosol sampler (Giano – Dado lab srl
Milano) equipped with $PM_{10}$ sampling head operating at constant air flow of $2.3m^3/h$ in accord with the European rule
EN12341.

Particle samples were collected on Teflon filters (PALL, Germany), 47 mm in diameter, with 2.0 μm nominal porosity and
99.5 % sampling efficiency for 0.3 μm aerodynamic particle diameter.

The filters were shipped to Italy, being kept at -20 °C, and stored in Petri plastic dishes, until they are cut, extracted and
analyzed.

In the laboratory the $PM_{10}$ mass was determined by weighting filters before and after the sampling on a 5-digit
microanalytical balance (Sartorius) equipped with an ionic cannon to avoid mass fluctuations due to the electrostatic charge
of filters.

Before weighting, filters were stored in a dryer for 48 h, with $50 \pm 5$ % relative humidity. A fourth of each filter was
devoted to the ion determination by ion chromatography (IC). The ¼ filter was extracted in ultrapure water (Resistivity > 18
MΩ) in an ultrasonic bath at room temperature, then the ionic content was determined by three ion chromatographs (two
ICS-1000 and one DX500 Thermo Fisher Scientific Inc., USA) equipped with Gilson 222 XL autosampler. This system
makes it possible to simultaneously determine both anions (inorganic and selected low molecular weight organic anions) and
cations within 10 min. Details on IC measurements are reported in (Becagli et al., 2022).

In order to exclude the sea salt contribution to the total $SO_4^{2-}$ budget, the non-sea salt (nss) $SO_4^{2-}$ was calculated as follows:

$nssSO_4^{2-} = SO_4^{2-} - (SO_4^{2-}/Na^+)_{sw} * Na^+$

were $SO_4^{2-}$ and $Na^+$ are the measured concentrations in the aerosol samples (as $ng/m^3$) and $(SO_4^{2-}/Na^+)_{sw}$ is the $SO_4^{2-}$ to $Na^+$
ratio in sea water 0.25 w/w (Henderson and Henderson, 2009).

## 2.3 Gaseous DMS sampling and analysis

DMS sampling is performed near the main MZS building at about 5 m distance from the sea at 2 m above sea level. The
sampling was done at sub daily resolution (typically 4 samples in 24 hours) by filling electropolished stainless steel canisters
by compressing the air at 4 bars within several minutes with a membrane pump (Millipore XX5522050). Before sampling,
the canister was filled and emptied two times with ambient air in order to wash the canister avoiding memory effects from
the previous samples.



DMS measurements were made in the MZS laboratories by using a gas chromatograph equipped with a flame photometric detector (HP6890, 393 nm). DMS is trapped in an ethanol bath at -70°C on a porous polymer resin based on 2.6-diphenilene oxide (Tenax®) contained in a sample loop. DMS is injected in the GC by thermal desorption in boiling water. Working

conditions are reported in detail by Legrand et al. (2001). Daily calibrations were achieved by using a permeation tube (VICI Metronics, Santa Clara California) thermostated at 30°C. The permeation tube was calibrated, and its stability was checked, resulting in less than 5% changes within one year (Preunkert et al., 2007). The detection limit is 0.2 ng, leading to an atmospheric detection limit of 0.12 pptV for 6 L of sampled air.

**2.4 DMSP in sea water sampling and analysis**

During both ACs sea water samples were collected at the sea surface in two sites about once a week, when the piers were free from sea ice and meteorological condition allowed to use a boat. The two sampling sites were chosen in ice free water about 2 miles from the coast and from the sea ice margin and 2 miles from each other. Few samples in November 2018 were collected in a hole in the pack ice, about 1 km from MZS in the Gerlache Inlet. The samples were collected in Schott® bottles, acidified to pH <2 by adding small amount of distilled $HNO_3$, then hermetically sealed, and maintained at 4°C until

the analysis. The analysis was accomplished at the Korea Polar Research Institute (KOPRI) laboratories. The preserved DMSP sample was hydrolyzed to gaseous DMS using 10 M NaOH (addition of 0.25 mL per mL sample) and was allowed to react overnight in the dark. Then, DMS was measured by using a gas chromatography equipped with pulsed flame photometric detector (GC-PFPD) as described in (Park et al., 2014). The DMSP sample was measured in duplicate, and the analytical precision was generally better than 5%.

**2.5 DMS UV-A and SW irradiance measurements**

Measurements of downwelling photosynthetically active radiation and shortwave irradiance were made at Icaro Camp throughout 2018-2019 and 2019-2020. Measurements of UV-A and UV-B irradiances were added during the 2019-2020. The shortwave irradiance was measured with a compact Kipp and Zonen Splite sensor, while the photosynthetically active radiation (PAR) with a Li-cor 190R. Both instruments were calibrated by the manufacturer before deployment. In additions,

the radiometers were installed at the Lampedusa Climate Observatory before deployments in Antarctica in 2018 and 2019, where they were compared with instruments continuously running at the site. The calibration of the shortwave irradiance radiometers at Lampedusa is traceable to the World Meteorological Organization World Radiation Reference scale (e.g., di Sarra et al., 2019). The PAR calibration scale has been maintained at Lampedusa relying on the initial manufacturer calibration and on the local calibration of a multi-band radiometer through the Langley plot method (Trisolino et al., 2017).

A UV-A and a UV-B broadband radiometer, a Delta-T UV2/ap and UV2/bp, respectively, were added for the 2019-2020 AC. The radiometers were calibrated at Lampedusa before the campaign by comparison with measurements of spectral irradiance performed with a double monochromator Brewer spectrometer. Nominal spectral response functions (peaked respectively at 373 ad 313 nm for the UV-A and UV-B, corresponding bandwidths of about 31 and 26 nm) were used in the



determination of the broadband calibration. The Brewer spectrophotometer is regularly calibrated on site with 1000 W FEL
lamps (Di Sarra et al., 2008).

## 2.6 Wind Speed and direction data

Wind speed and direction data were measured at Eneide automatic weather station (AWS) (74° 41' 45"S 164° 5' 32"E) that
takes part of the Meteo-Climatological Observatory at MZS and Victoria Land maintained by the Italian National
Programme of Antarctic Research (http://www.climantartide.it). This AWS is nearest to both the DMS and aerosol sampling
sites and it have been used in this work.

## 2.7 Satellite data of sea ice and Chlorophyll

Daily maps of southern hemisphere sea ice cover were obtained from the National Snow & Ice Data Center. Information on
sea ice extent is derived from the analysis of satellite passive microwave brightness temperature data from the Nimbus-7
Scanning Multichannel Microwave Radiometer (SMMR) and from a series of Special Sensor Microwave Imager (SSM/I)
and Special Sensor Microwave Imager/Sounder (SSMIS) instruments (Fetterer et al., 2017). The nominal spatial resolution is
$25 \times 25$ km$^2$. Data were downloaded from https://nsidc.org/data (last accessed 2021-12-22).
Satellite-derived daily L3 data sets of surface Chlorophyll-a concentration with a 4km spatial resolution from the European
Space Agency's GlobColour Project (http://hermes.acri.fr) were obtained from the Copernicus Marine Environment
Monitoring Service (CMEMS, https://marine.copernicus.eu/). The Chl-a product is derived by reprocessing the merged ob-
servations from five satellite radiometers (MODIS on Aqua, VIIRS from Suomi-NPP and JPSS-1, and OLCI from Sentinel
3a and 3b). The GlobColour dataset is a common and appropriate choice for phytoplankton dynamics studies even in the
Southern Ocean (Ardyna et al., 2017; Cole et al., 2015).

## 2.8 Backward trajectories calculation

Ten-day back trajectories are calculated with the HYSPLIT-model (Stein et al., 2015). We used the ensemble method of the
model that has been incorporated directly into the code so that trajectories are automatically computed about a 3-dimensional
cube about the starting point at 300 m above MZS. The initial positions are not offset, just the meteorological data point
associated with each particular trajectory, so that all trajectories start from the same point
(https://www.ready.noaa.gov/documents/Tutorial/html/traj_ensem.html, last accesses 2021-12-22). The trajectories are
based on the meteorological fields of the Global Data Assimilation System (GDAS1) provided by the US National Weather
Service's National Centers for Environmental Prediction (NCEP) at one-degree resolution
(https://www.ready.noaa.gov/gdas1.php, last accesses 2021-12-22).





# 3. Result and discussion

Fig. 2 shows the time series of DMS, MSA and nssSO$_4^{2-}$ concentrations during the two summer Antarctic campaigns. In order to derive the stoichiometric ratio between the two aerosol species (MSA and nssSO$_4^{2-}$) and its gaseous precursor (DMS) the concentrations are reported in nMol/m$^3$.

**Figure 2. Time series of DMS, MSA and nssSO$_4^{2-}$ in the two AC: 2018-19 (plots on the left) and 2019-20 (plot on the right).**

The three compounds display very different pattern in the two summer ACs, as concerns timing and concentration maxima. During the 2018-19 AC the DMS concentration maxima are lower but last longer than in the 2019-20 campaign. The median and 75° percentile of the DMS concentration measured in the two ACs are 0.67 and 1.81 nMol/m$^3$, respectively. DMS

values higher than the 75° percentile occurred during 27%, and 17% of the time for the 2018-19 AC and 2019-20 AC,



respectively, although the maximum DMS concentration was lower (25.4 nMol/m$^3$) in the 2018-19 AC than in the 2019-20 AC (37.7 nMol/m$^3$). Basic statistics on the measured values of DMS, MSA, and nssSO$_4^{2-}$ in the two ACs are reported in Table 2, and are compared with measurements performed at other Antarctic sites and over the Southern Ocean. Biogenic aerosol data from Antarctic sites and the Southern Ocean are scarce, and even more so for measurements of DMS. However,

despite a general spatial (site to site) and temporal (year to year at the same site) variability some considerations can be made.

Despite its more southern position, the maximum at MZS occurs earlier (December) than at DDU and over the Southern Ocean (January). This is likely due to the influence of the polynya, where an early phytoplanktonic bloom, and consequently an early release of DMS to the atmosphere from the ice-free area, takes place. However, notwithstanding the large year to

year variability, the DMS summer mean and maximum concentration at MZS are lower than at the other sites reported in Table 1.

Maximum concentrations of MSA and nssSO$_4^{2-}$ at MZS occur in December (in correspondence with the DMS maxima) or in January.

The MSA and nssSO$_4^{2-}$ peaks coincide largely with those of its precursor DMS. However, the DMS number of moles is

always larger than that of MSA and nssSO$_4^{2-}$. The DMS life time in summer in Antarctica is modelled to be in the range 0.5-3 days (Faloona, 2009; Hezel et al., 2011). Consequently, we assume that the measured DMS comes from the open ocean areas near the sampling site and is not yet fully oxidized.

Although MSA concentrations show a high site-to-site variability, it appears that at MZS, MSA concentrations are higher than at the other sites reported in Table 1. Non sea salt SO$_4^{2-}$ values at MZS are of the same order of magnitude as at the

other Antarctic sites, except at Halley and on the ocean cruise, where smaller concentrations are found.

**Table 1.** Mean and standard deviation of DMS, MSA and nssSO$_4^{2-}$ concentrations in coastal Antarctic sites. § summer mean calculated over the years 1999-2003; * summer mean calculated over the years 1997/98, 2001/02, 2002/03; ° summer mean calculated over the years 1996/97, 1999/00, 2001/02.


| Station | Coord. | Year | Month | DMS (gas phase) | | MSA (aerosol) | | nssSO$_4^{2-}$ (aerosol) | | Reference |
|---|---|---|---|---|---|---|---|---|---|---|
| | | | | Mean (std dev) nMol/m$^3$ | Max nMol/m$^3$ | Mean (std dev) nMol/m$^3$ | Max nMol/m$^3$ | Mean (std dev) nMol/m$^3$ | Max nMol/m$^3$ | |
| **MZS** | 74°42' S, 164°07' E | 2018-19 | Dec | 2.32 (3.87) | 25.4 | 0.60 (0.65) | 3.41 | 2.02 (0.93) | 5.41 | this work |
| **MZS** | 74°42' S, 164°07' E | | Jan | 1.78 (1.88) | 10.7 | 1.47 (1.53) | 8.4 | 3.41 (1.75) | 8.90 | this work |
| **MZS** | 74°42' S, | 2019- | Dec | 2.90 | 37.7 | 1.50 | 8.29 | 3.19 | 11.3 | this work |



| | | | | | | | | | | |
|---|---|---|---|---|---|---|---|---|---|---|
| | 164°07' E | 20 | | (5.98) | | (1.88) | | (2.58) | | |
| MZS | 74°42' S, 164°07' E | | Jan | 0.61 (0.49) | 2.7 | 0.51 (0.50) | 2.58 | 2.46 (1.39) | 7.55 | this work |
| DDU | 66°40'S, 140°01'E | 1998-99 | Jan | 13.10 (6.10) | | 0.60 (0.30) | | 3.80 (1.4) | | (Jourdain and Legrand, 2001) |
| Neumayer | | 1983-95 | Jan | | | 1.60 (0.80) | | 3.95 (1.39) | | (Minikin et al., 1998) |
| DDU | 66°40'S, 140°01'E | 1991-95 | Jan | | | 0.66 (0.20) | | 3.57 (0.41) | | (Minikin et al., 1998) |
| Halley | 73°35'S 26°19'W | 1991-93 | Jan | | | 0.35 (0.21) | | 0.93 (0.37) | | (Minikin et al., 1998) |
| DDU | 66°40'S, 140°01'E | 1991-2003 | summer | 8.2§ | 227.2§ | 0.36* 0.76° | | 1.89* 2.57° | | (Preunkert et al., 2007) |
| Palmer | 67.77S, 64.05W | 1994 | Jan-Feb | 4.92 (3.89) | | 1.76 (1.42) | | 2.85 (2.24) | | Barrensheim et al., 1998 |
| Halley | 73°35'S 26°19'W | 2004 | Jan-Feb | | | 0.87 | | | | (Read et al., 2008) |
| Halley | 73°35'S 26°19'W | 2005 | Jan-Feb | | | 1.47 | | | | (Read et al., 2008) |
| Zhongshan | 69°22'S 76°22'E | 2005-2008 | Jan-Mar | | | 0.46-0.87 | | 1.17-2.43 | | (Zhang et al., 2015) |
| Cruise in Southern Ocean | 40°S-76°S; 170°E - 110°W | 2018 | Feb-Mar | 36.8 (39.2) | 445.4 | 0.32 | | 1.54 (0.34) | | (Yan et al., 2020) |
| King Sejong | 62.2° S, 58.8° W | 2018-2020 | Dec-Mar | 1.79 (1.50) | 24.3 | 1.00 (0.34) | 2.05 | 2.35 (1.01) | 5.82 | (Jang et al., 2022) |

**3.1 Factors controlling atmospheric DMS: wind speed and direction, Chl and sea ice extent**

The difference between the DMS evolution in ACs 2018-2019 and 2019-2020 shown in fig. 2 may be due to different

causes: biological (DMS production from phytoplankton), physical (DMS flux from sea water to the atmosphere), or chemical (different rates of DMS oxidation in the atmosphere).

The role of wind speed has been firstly investigated.



The DMS sea air flux depends on oceanic DMS concentrations and on the transfer velocity coefficient ($k_w$), which is a strong nonlinear function of wind speed (Nightingale et al., 2000; Vlahos and Monahan, 2009). Moreover, the measured DMS atmospheric concentrations at coastal sites are expected to be heavily influenced by the wind direction, as DMS sea-to-air transfer occurs only at the ice-free ocean surface, and air masses coming from the ice sheet do not contain DMS.

In the AWS hourly data set wind coming from the continental ice sheet, i.e., wind direction between 200 and 350°, were identified in the period 6 December-6 January when maximum DMS release in the atmosphere is measured in both campaigns. Surprising, the fraction of time with winds blowing from the ice-sheet is higher in 6 Dec. 2018 - 6 Jan.2019 (when more samplings with DMS concentration > 75th percentile are observed) than 6 Dec. 2019 - 6 Jan. 2020 (68% and 48% of time respectively, see Table 2), suggesting that the different evolution of the DMS concentration is not produced by a different pattern of wind direction.

Data display conversely a dependency on wind speed. The distribution of wind speeds for cases with high DMS concentrations (> 75° percentile) displays that the 82-83% of wind speeds are in the range 1-10 m/sec in the two campaigns. This is consistent with modelized DMS transfer rate as function of wind speed that increases as wind speed increases up to 10 m/sec but decreases as wind speed increases (Vlahos and Monahan, 2009).

If we consider the wind speed velocity in the range 1-10 m/sec as the best conditions for DMS transfer rate, by looking at the wind distribution in the above mentioned time period for the two year we can see an opposite pattern than expected with lower percentage of favorable wind speed in the year with higher number of DMS concentration higher than 75th percentile (table 2).

**Table 2**. Number of data and percentage of DMS concentration higher than 75th percentile, wind direction from the ice sheet and wind speed lower than 10 m/sec.

| | DMS conc. >75th percentile | | Wind Direction from the ice sheet | | Wind speed <10 m/sec | |
|---|---|---|---|---|---|---|
| | n. of data | % | n. of data | % | n. of data | % |
| **6 Dec. 2018-6 Jan.2019** | 63 | 42 | 525 | 68 | 425 | 55 |
| **6 Dec. 2019-6 Jan.2020** | 49 | 38 | 366 | 48 | 569 | 74 |

Therefore, large differences in the DMS sea to air transfer velocity coefficient ($k_w$) do not seem to occur, and are not expected to be the cause of the different behavior of DMS evolution in the two ACs. This result is consistent with previous modeling studies(Sciare et al., 2000) that, based on a 3-D chemistry-transport model over 10 years at Amsterdam Island, found that the large year-to-year variability of the seasonal atmospheric DMS cycle can not be explained by changes of meteorological processes controlling the $k_w$ factor or by changes of atmospheric oxidants, but most likely by changes in oceanic DMS concentrations. Similarly, another study estimated, on the basis of 15-year meteorological data, that interannual variations of wind speed and sea surface temperature induce changes of less than 10% in the DMS sea air flux


(Kettle and Andreae, 2000). They suggested that greater interannual sea air flux variations are caused by variations of seawater concentrations of DMS.

Therefore, the DMS concentration in the atmosphere does not seem to be controlled by physical processes at the ocean/atmosphere interface but is likely related to biological processes in the ocean.

Previous studies found that DMS concentrations in seawater are related to phytoplanktonic biomass (expressed by Chl-a concentration) or to primary productivity, that are in turn controlling the biogenic aerosol (Minikin et al., 1998; Preunkert et al., 2007).

Some of the highest DMS concentrations in seawater worldwide (>300 nMol/L) have been reported from the Ross Sea, Antarctica, associated with seasonal blooms of the phytoplankton *Phaeocystis a.* (Ditullio et al., 2003; Gambaro et al., 2004), a high-DMSP producer (Liss et al., 1994).

Fig. 3 shows the time series of measured atmospheric DMS and marine DMSP concentrations, and of satellite-derived Chl-a in the two ACs. The satellite determinations of Chl-a are averaged over the region 162.5977°E, 77.666°S; 171.2109°E,

72.2168°S, which corresponds to the area of a rectangle 600x300 km covering the polynya area facing MZS. Evidently, the first maxima in Chl-a are followed by increased DMS values in the atmosphere. Also, the first Chl-a double-peak in late November early December 2019 is higher (1.2-1.8 mg/m$^3$) than the first peak in mid-December of 2018 (0.9 mg/m$^3$). A similar pattern is visible in DMS concentration, with a higher peak in early December 2019 (921 pptV = 37.7 nMol/m$^3$) than in late December 2018 (620 pptV = 25.4 nMol/m$^3$). This seasonal DMS accumulation may be caused by a combination of

factors, including high DMS production rates, limitation of bacterial DMS consumption at low temperatures, and saturation of biological DMS consumption rates (Ditullio et al., 2003; del Valle et al., 2009).





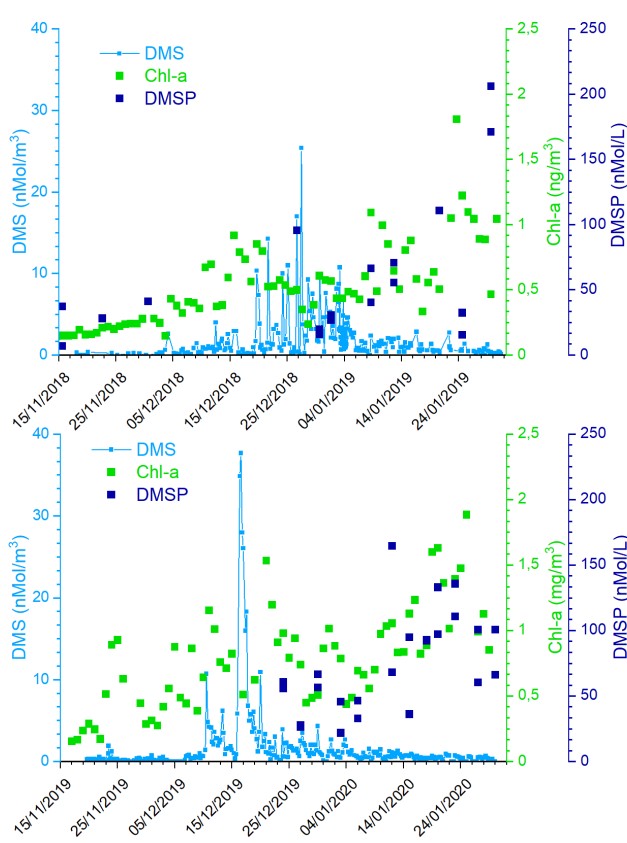

**Figure 3. Time series of measured atmospheric DMS, DMSP in sea water, and satellite-derived Chl-a (daily, area averaged concentration in the region 162.5977°E, 77.666°S; 171.2109°E, 72.2168°S, corresponding to a rectangle 600 x 300km) in the two ACs.**

Particularly relevant is the time lag of about 15 days in both Antarctic campaigns between the Chl-a and the DMS peaks. This delay is expected because DMS emissions depend on the physiological state of the phytoplankton. In particular, large emissions are connected with the phytoplanktonic senescent phase associated with stress factors, such as increasing solar radiation due to the shallowing of the depth of the mixing layer (Simó et al., 1999; Vallina and Simó, 2007), consumption of nutrients (Sunda et al., 2007; Zindler et al., 2014), grazing (Savoca and Nevitt, 2014), and bacterial decomposition (Kiene and Bates, 1990; Lomans et al., 2002). The senescent phase, and the DMS emission, generally follows the maximum of phytoplankton biomass coincident with the peak of Chl-a concentration.

It is interesting to note the presence in both the ACs of a second Chl-a peak in January. The Chl-a concentration peak in January is higher than the one in December, and it is not associated with an increase in DMS concentrations (cf. Fig. 3),



despite the high concentration of DMSP in surface seawater. This is probably due to the difference in community composition, which is dominated by *Phaeocystis a.* in early summer (November-December) when sea ice melts, and later by diatoms (Bolinesi et al., 2020; Innamorati et al., 2000). Indeed, it has been shown that in the Ross Sea the DMS:chl-a ratio (58–78 nMol/µg) was significantly higher in waters dominated by *P. antarctica* compared to diatom-dominated waters (2–12 nMol/µg) (DiTullio and Smith, 1995). Besides, (Del Valle et al., 2009) by investigating the pattern of biological DMS

consumption, found that while it remained relatively low and constant throughout the spring (0.05–0.21 d$^{-1}$), and higher (0.22–0.98 d$^{-1}$; i.e., faster biological turnover) in summer. The spring slow biological turnover probably contributed to the DMS buildup during the early bloom, while the fast biological turnover helped in producing low DMS concentrations in summer (3.2–16.8 nMol/L).

The higher biological DMS consumption in January than in December can explain the apparent anomaly of the higher

concentration of DMSP in near surface sea water and low DMS in the atmosphere. Unfortunately, we could not collect sea water samples during the period of maximum atmospheric DMS concentration due to lack of safe conditions which prevented to reach the open sea.

In addition to the different levels of Chl-a and DMS during the two Antarctic campaigns, the Chl-a peaks occurred with a different timing. In particular, in summer 2018-19 Chl-a peak occurs later than in the 2019-2020 summer.


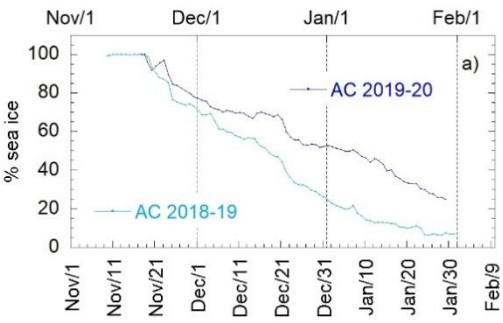

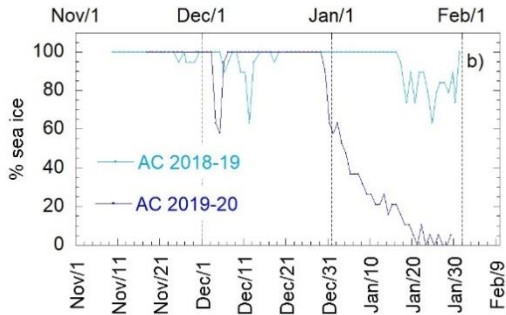

**Figure 4. Percentage of the area covered by sea ice for the (a) Ross Sea (800x800 km$^2$) and (b) the restricted area of polynya in the Ross Sea (100x100 km$^2$) for the two ACs.**



It is well known that phytoplanktonic blooms of *Phaeocystis a.* occur at the beginning of sea ice melting (Arrigo et al., 1998; Stefels et al., 2018). Fig. 4 shows the sea ice coverage, as determined by satellite observations, in two areas of the Ross Sea, for 2018-19 and 2019-20. When the whole Ross Sea area is considered (an area about 800x800 km$^2$), it appears that the ice cover is higher in 2019-20 than in 2018-19, which seems in contrast to the timing of Chl-a increases. However, analyzing in detail the area of the polynya facing the sampling site (about 100x100 km$^2$ wide), we may observe that sea ice starts melting

earlier and decreases faster in 2019-20 than the previous year. This evolution of sea ice in the region surrounding MZS could produce a shorter but more intense phytoplanktonic bloom in the polynya area in 2019-2020, and seems to be consistent with the observed evolution of DMS and related parameters. This suggests that the polynya areas close to MZS play a dominant role for the phytoplanktonic cycle and production of biogenic aerosol precursors.

### 3.2 The relationship between DMS and its oxidation products (MSA and nssSO$_4^{2-}$): a schematic representation of mechanisms

The simultaneous measurements of atmospheric DMS and its oxidation products allow to study the dynamic processes occurring in the atmosphere leading to the formation of biogenic particulate matter. In this section we discuss the overall

evolution of DMS, MSA and nssSO$_4^{2-}$ and their relationships, with the aim of identifying periods for which occurring processes, with respect to the DMS oxidation pathways, can be determined. Specific periods in which near/far sources of the different compounds and oxidation processes for different DMS emission conditions, are identified and discussed as explanatory cases.

Fig. 2 shows that in both ACs MSA displays a time evolution similar to nssSO$_4^{2-}$, with simultaneous peaks. Conversely, the

time evolution of MSA and nssSO$_4^{2-}$ differs from that of DMS in the two ACs: (i) during the 2019-2020 AC, maxima of biogenic aerosol compounds occur with a short time difference (24 h) with respect to the DMS peaks, (ii) during the 2018-2019 AC, the largest MSA and nssSO$_4^{2-}$ peak occurs one month later than DMS.

In the period 15-18 December 2019 (Fig. 5) MSA and nssSO$_4^{2-}$ maxima are associated with DMS. This case offers an exceptional example to understand the dynamics of biogenic aerosol formation. In this period air masses arrived from the

area of the Ross Sea surrounding the sampling site (Fig. 6a) passing at low height on the near sea areas not covered by sea ice, therefore in correspondence with strong DMS emission from sea water. The wind speed is almost constant at about 10 m/sec (which, as discussed above, is most favorable for the DMS transfer to the atmosphere). The measured relative humidity is 100%, and UV radiation is attenuated by clouds until 17 December at 00:00 (Fig. 5). In these conditions, the DMS concentration reached a maximum of 32.8 nMol/m$^3$ on 16 December (average over the period 9:00-21:00 LT).

Assuming that DMS emission from the ocean remains constant also in the following day, due to the constant wind speed and direction (Fig. 5), we expect that the UV radiation increase occurring in the following 24 h stimulated the DMS oxidation processes, leading to the MSA and nssSO$_4^{2-}$ formation and to a decrease of atmospheric DMS concentrations. MSA and nssSO$_4^{2-}$ reached the maximum concentration on December 17$^{th}$ (8.3 and 9.9 nMol/m$^3$ for MSA and nssSO$_4^{2-}$, respectively,





for the sampling time 9:00-21:00 LT), when the DMS concentration was 12.6 nMol/m³. Within the approximation of
constant DMS emissions, the amount of lost DMS (32.8 - 12.6=20.2 nMol/m³) equals the amount of formed MSA + nssSO$_4^{2-}$
(8.3+9.9= 18.2 nMol/m³). In these conditions, the MSA/nssSO$_4^{2-}$ ratio ranges from 0.68 to 0.94 mol/mol, indicating a high
branch ratio between OH addition (leading to MSA formation) and abstraction (leading to nssSO$_4^{2-}$ formation) reactions.

**Figure 5. DMS, MSA and nssSO$_4^{2-}$, wind speed and direction, relative humidity (RH) and UVB for the time 15-19 December 2019**
**and 9-14 December 2019. DMS and UV-B data are averaged over the time interval of the corresponding aerosol sampling (12h).**




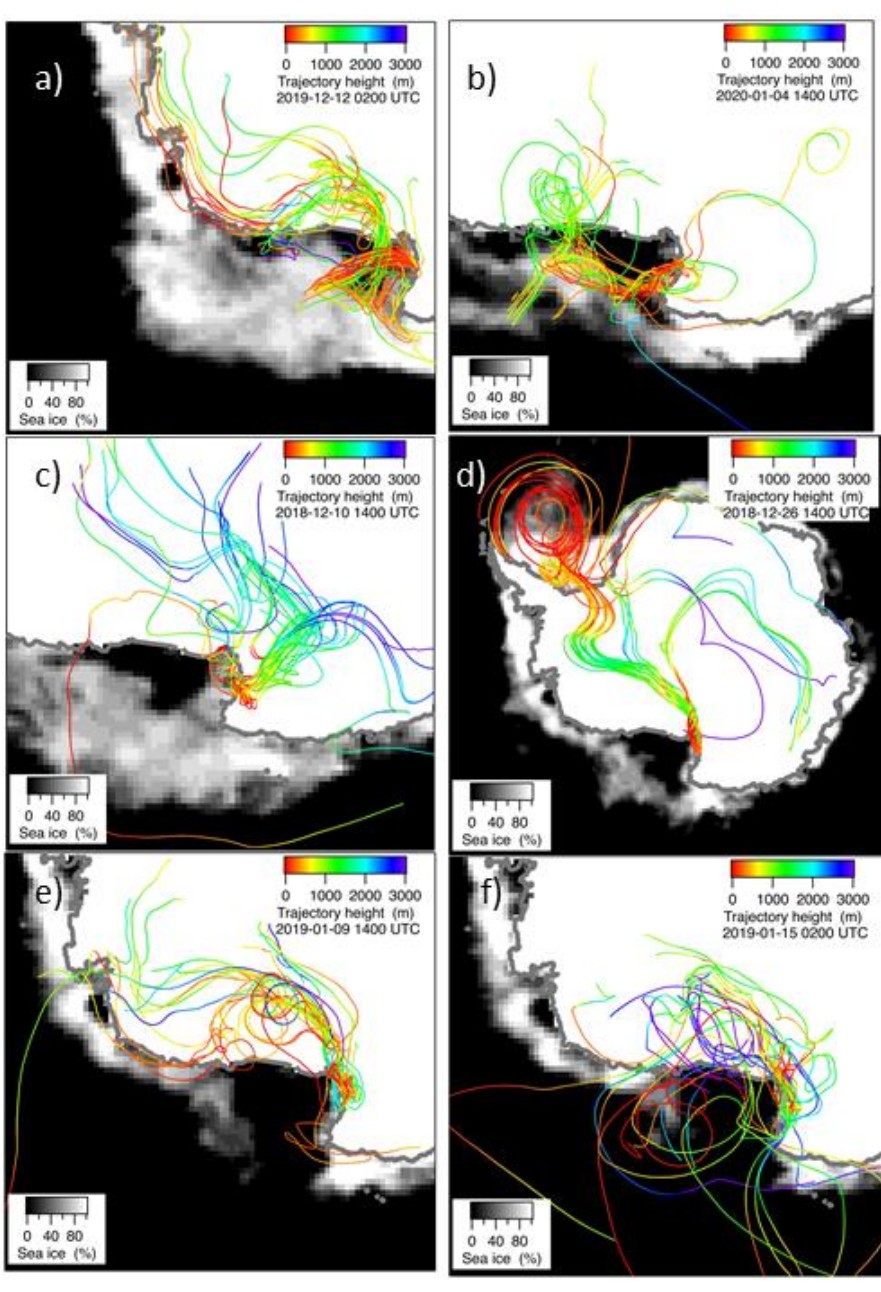

**Figure 6. Ensemble trajectories at 300 m a.g.l. arrival height for the days 16 December 2019 (a), 4 January 2020 (b), 10 December 2018(c), 26 December 2018 (d), 9 January 2019 (e), 15 January 2019 (f) together with the ice cover for these days. Trajectory**
**height along the route and sea ice percentage are presented as color or gray scales respectively.**





A similar situation occurred also in the period 9-15 December, 2019, and 3-7 January, 2020 (the latter reported as an example in Fig. 5, and the corresponding backward trajectories in Fig. 6b) even though with different intensity of DMS emissions.

In the 2018-2019 AC, conversely, MSA and $nssSO_4^{2-}$ maxima do not strictly coincide with high DMS concentrations. In particular:

1. The first peaks of MSA and $nssSO_4^{2-}$ occur on 10-11 December, 2018, about 10 days before the main DMS peak. Sea ice is still present near the sampling site and DMS can not escape locally to the atmosphere. Therefore, its concentration in the atmosphere is very low. At this time MSA and $nssSO_4^{2-}$ were very likely transported from areas far from the sampling site, where an early phytoplankton bloom was taking place. Backward trajectories show air masses coming from the ice sheet (cf. Fig. 6c); therefore, we can suppose that MSA and $nssSO_4^{2-}$ came from oceanic sectors far from the sampling site. In this period, the MSA/$nssSO_4^{2-}$ ratio was quite low (about 0.2), indicating generally aged air masses.

2. In the period 19-30 December, 2018, several DMS peaks were measured, associated with low or moderate MSA and $nssSO_4^{2-}$ concentrations. In this period DMS emissions from the ocean are expected to be high and to be captured by the air masses traveling over the sea. This period is characterized by high wind speeds; DMS concentration spikes are higher when wind speed drops (Fig. 7). The high and variable wind speed does not allow the MSA and $nssSO_4^{2-}$ formation and accumulation in the air mass travelling from the DMS near source area to the sampling site. The backward trajectories show that airmasses came from the Weddel Sea, far from the sampling site (Fig. 6d), crossing the ice sheet and passing at low elevation over the Terra Nova Bay polynya just before arriving to the sampling site. The presence of generally aged air masses in this period is confirmed by the low MSA/$nssSO_4^{2-}$ ratio (0.35 on average). Therefore, in these periods airmasses containing MSA and $nssSO_4^{2-}$ came from an oceanic sector far from the sampling site, whereas the DMS enters the air mass over the polynya just before the sampling site.

3. In the two periods 9-13, and 15-18 January, 2019, high MSA and $nssSO_4^{2-}$ concentrations ($> 5$ nMol/m$^3$) are measured, while DMS concentrations are very low ($< 3$ nMol/m$^3$). In these periods wind speed was very low ($<$ 5m/sec) (Fig. 7). Backward trajectories show air masses coming from the ice sheet for the period 9-13 January (Fig. 6e) and routes extremely variable sometimes coming from the northern part of Ross Sea uncovered by ice at this time. (Fig. 6f). Indeed, the MSA/$nssSO_4^{2-}$ ratio, is quite low on 9-13 January (0.40 Mol/Mol) when air masses coms from ice sheet and therefore far from the sampling site, while it reaches the highest measured value (up to 0.96 Mol/Mol) during 15-18 January suggesting the presence of freshly formed biogenic aerosol from DMS formed in the northernly in the Ross Sea.



430

**Figure 7. DMS, MSA and nssSO₄²⁻, wind speed and direction, relative humidity (RH) and short-wave irradiance for the time 19-30ᵗʰ December 2018 and 9-20ᵗʰ January 2019. DMS and SW irradiance data are averaged over the same time interval of the aerosol sampling (12h).**





### 3.3 Quantification of the biogenic aerosol contribution to PM$_{10}$

In the previous section we highlighted the variability of the MSA/nssSO$_4^{2-}$ ratio as a function of the air masses aging. In order to find a characteristic branch ratio for the DMS oxidation, the MSA concentrations are reported versus nssSO$_4^{2-}$ concentrations in Fig. 8. A somewhat different pattern of MSA concentration with respect to nssSO$_4^{2-}$ concentrations appears. We separated data in two classes, respectively for nssSO$_4^{2-}$ values lower and higher than the somewhat arbitrary value of 3 nMol/m$^3$. The slope of the MSA-nssSO$_4^{2-}$ relationship appears to change approximately around this value. Small changes in the results that will be discussed occur when a different threshold value in the range 2.5-4 nMol/m$^3$ is chosen.

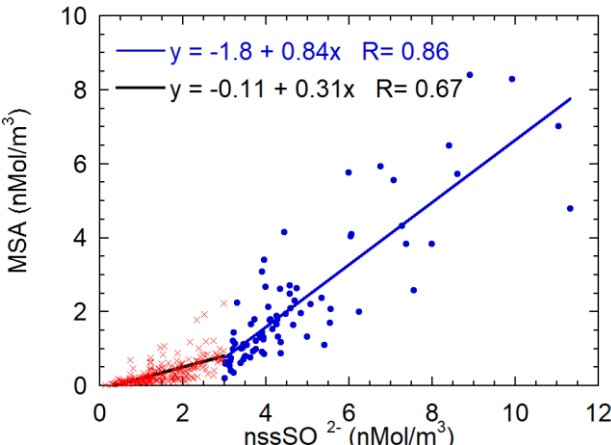

**Figure 8. Scatter plot of MSA concentrations versus nssSO$_4^{2-}$. The two regression lines are calculated for nssSO$_4^{2-}$ concentration lower (red crosses, black line) and higher (blue dots and line) than 3 nMol/m$^3$.**

MSA shows a relatively high correlation with nssSO$_4^{2-}$ for both classes (even though the correlation is worse for low nssSO$_4^{2-}$ concentrations) suggesting that both species have a common source, as expected for DMS oxidation. The slope of the regression line is higher for the class with nssSO$_4^{2-}$ > 3 nMol/m$^3$. The two different slopes can be associated with different situations: for nssSO$_4^{2-}$-concentrations below 3 nMol/m$^3$, the MSA-nssSO$_4^{2-}$ relationship is expected to be produced by aged biogenic aerosol, with possible additional nssSO$_4^{2-}$ contributions coming from the oxidation of SO$_2$ emitted by the near volcano Erebus (Boichu et al., 2015)( Fig. 1), or from long range transport from northern latitudes (Minikin et al., 1998).

Conversely, nssSO$_4^{2-}$ values higher than 3 nMol/m$^3$ appear to be generally associated with the presence of freshly formed biogenic aerosol. The abscissa-intercept of the regression line is 2.1 nMol/m$^3$, corresponding to 202 ng/m$^3$. Once this background contribution is subtracted, the MSA/nssSO$_4^{2-}$ derived from the regression line is 0.84±0.06 Mol/Mol (that is the





same value expressed as w/w as MSA and $SO_4^{2-}$ have almost the same molar mass). We assume that this value can be considered as the mean branch ratio between the two species in the newly formed biogenic aerosol in summer at this high southern latitude. Several other studies report quite different MSA/nss$SO_4^{2-}$ ratios, both in aerosol and snow layers (Becagli et al., 2005; Legrand and Pasteur, 1998; Minikin et al., 1998; Mulvaney and Wolff, 1994; Preunkert et al., 2008; Zhang et al., 2015).However, some of these determinations are affected by fractionation effects in the aerosol during the transport from source regions to the sampling site, and by different deposition processes. In this study, thanks to the closeness and strength of the DMS source during periods of ice-free polynya in the short range of the sampling site, and to the opportunity to find air masses containing DMS and both its freshly formed oxidation products, we believe that it is possible to obtain a reliable MSA/nss$SO_4^{2-}$ branch ratio.

Considering the samples when nss$SO_4^{2-}$ is higher than 3 nMol/m$^3$ as representative of the presence of freshly formed biogenic aerosol, is it thus possible to quantify the role played by newly formed biogenic aerosol on the total PM$_{10}$ mass. On average, the sum of biogenic nss$SO_4^{2-}$ (i.e. by subtracting the nss$SO_4^{2-}$-background) and MSA accounts for 17% of the PM$_{10}$ mass, with maxima in single samples as high as 56%. This contribution is relevant, and its quantification is important also from a climatic point of view, as biogenic aerosol can constitute an important source of cloud condensation nuclei over the Southern Ocean.

## 4. Summary and conclusions

Simultaneous high time resolution measurements of sulfur-compounds have been collected at a costal Antarctic site (MZS) during two summer campaigns (2018-2019 and 2019-2020), to provide information on marine biological activity in the nearby polynya in the Ross Sea and on the influence of biogenic and atmospheric processes on biogenic aerosol formation.

Data on atmospheric DMS concentration are scarce especially in Antarctica. The DMS-maximum at MZS occurs in December, one month earlier than at the other sites at lower southern latitudes where measurements are available. The maximum of DMS concentration appears to be connected with the phytoplanktonic senescent phase following the bloom of *P. antartica* that occurs in the polynya area closest to the sampling site, when sea ice opens up.

The second plankton bloom is related to diatoms and occurs in January. During this bloom, despite the high DMSP concentration in sea water, atmospheric DMS remained low probably due to its fast biological turnover in sea water in this period (Del Valle et al.,2009).

The intensity and timing of the DMS evolution during the two years also suggests that only the portion of the polynya close to the sampling site produces a discernible effect on the measured DMS.

Several studies highlight the necessity to determine the branch ratio between MSA and nss$SO_4^{2-}$ from DMS oxidation at high latitudes. However, the DMS oxidation responds to multiple processes and controllers, including concentration of atmospheric oxidants and meteorological factors; therefore, the values of the branch ratio found in the literature vary considerably (e.g., Bates et al., 1992, Preunkert et al., 2008; Yan et al., 2020). In this study, the closeness to the DMS source



area, and the occurrence of air masses containing DMS and freshly formed oxidation products allow a reliable derivation of the branch ratio. The MSA/nssSO$_4^{2-}$ branch ratio for newly formed biogenic aerosol is estimated to be 0.84 ±0.06.

Conversely, data suggest that for aged airmasses with low DMS content, an enrichment of nssSO$_4^{2-}$ with respect to MSA, due to the presence of background concentration of nssSO$_4^{2-}$ from volcanic origin (Erebus) or from long range transport, takes place. Therefore, the aged air mass presents MSA/nssSO$_4^{2-}$ratio lower than in newly formed biogenic aerosol.

By considering the sum of MSA and biogenic nssSO$_4^{2-}$ in periods impacted by fresh biogenic aerosol, we estimate that the mean contribution of biogenic particulate matter to PM$_{10}$ is 17%, with a maximum of 56%. The high contribution of biogenic aerosol to the total PM$_{10}$ mass in summer in this area highlights the dominant role of the polynya area on biogenic aerosol formation. This is especially important due to the possible relevant role played by this aerosol in CCN formation.

Finally, due to the regional and year-to year variability of DMS and related biogenic aerosol formation, we stress the need of

long-term measurements of atmospheric DMS and biogenic aerosol along the Antarctic coast and in the Southern Ocean. This is particularly important in this phase, in which increasing temperatures and fast changes of ice distribution and properties are expected to affect other environmental parameters, such as primary productivity, formation of biogenic aerosols, and consequent climate-related parameters.


**Acknowledgements**

The research was financially supported by the MIUR (Italian Ministry of University and Research) and PNRA (Programma Nazionale di Ricerca in Antartide) through the PNRA16_00065-A1 Project "Correlation between biogenic aerosol and primary production in the Ross Sea -BioAPRoS".

Meteorological data are furnished by the "Meteo-Climatological Observatory at MZS and Victoria Land" project funded by the PNRA (PNRA14_00019) and managed by staff of the Italian National Agency for New Technologies, Energy and Sustainable Economic Development (ENEA) (www.climantartide.it)

We would like to thank the National Snow and Ice Data Center (NSIDC) funded through the NASA Earth Science Data and Information System (ESDIS) project and the Global Data Assimilation System (GDAS1) provided by the US National

Weather Service's National Centers for Environmental Prediction (NCEP) for archiving and publishing the data.

Finally, a special thanks to the logistic and scientific staff at "Mario Zucchelli Station" in the Antarctic campaigns 2018-19 and 2019-20, without them this work could not have been accomplished.




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

**Code/Data availability**

Data are available on request to the corresponding authors.


**Author contribution**

S.B.: Conceptualization; Investigation; Writing. E.B., Methodology; Data curation. S.Bo.: Methodology; Data curation. LC: Methodology; Data curation. AdS: Conceptualization; Investigation. M.F: Methodology; Data curation. P.G.: Methodology; Data curation. J.H.: Conceptualization; Investigation. L.L.: Conceptualization; Investigation. M.L.: Investigation. A.M.:

Methodology; Data curation. M.M. Methodology; Data curation. C.M.: Methodology; Data curation. D.M.: Conceptualization; Investigation.   C.N.: Methodology; Data curation. G.P.: Methodology; Data curation. K.-T.P. Methodology; Data curation. S.P.:Investigation. M.S. Validation; Investigation. M.V.:Methodology; Data curation. R.Z.: Methodology; Data curation. R.T.: Validation; Investigation; Funding acquisition.


**Declaration of interests**

The authors declare that they have no known competing financial interests or personal relationships that could have appeared to influence the work reported in this paper.