# Peer review of "Factors controlling atmospheric DMS and its oxidation products (MSA and nssSO42-) in the aerosol at Terra Nova Bay, Antarctica"

_Atmospheric Chemistry and Physics, 2022_

## Referee Comment (RC1)

**Referee Report**

Factors controlling atmospheric DMS and its oxidation products (MSA and nssSO$_4$) in the aerosol at Terra Nova Bay, Antarctica

by

Silvia Becagli et al.

This is an comprehensive report on two austral summer field campaigns that examined the biogenic compound DMS, and its oxidation by-products MSA and nssSO$_4$- at a coastal Antarctic site ( Northern Victoria Land ), in the vicinity of a polynya in the Ross Sea. The atmospheric measurements are augmented by seawater data on DMSP the precursor to DMS, together with various phytoplankton-related biological processes and parameters.

The manuscript treats an important topic as data on these compounds is quite limited in the Southern Ocean and Antarctic. The influence of the nearby polynya is also noteworthy. One aspect I think that could be improved is the relationship of aerosol concentration with sea ice change. Although the data on sea ice evolution is presented it is not discussed much in relation to the evolution of aerosols at the sampling sites. In Lines 405-409 the authors suggest the high MSA concentrations that preceded the DMS and phytoplankton peaks and may be due to long-range transport. For an alternative view which involves the release of these sulfur compounds during the period of sea ice melt,  please see the recent analyses in the SO  by Gabric et al. 2005, GBC v19, and in the Arctic by Gabric et al 2018 BAMS, v99.

The overall presentation is well structured and clear, and except for some minor issues the language is quite good. The document seems to have too many paragraphs which may be a formatting issue which arose in conversion to PDF --not sure.

However, the manuscript could be improved by a careful edit, and the authors should check their use of prepositions (English usage is annoyingly different to Italian here), some examples being :-

"presents the results on…" should be, "presents the results of…"

"DMS is formed in the breakdown…" should read, "DMS is formed by the breakdown.."

"the need of long-term measurements.." should read, " the need for long-term measurements"

"We separated data in two classes.." should read, "We separated data into two classes"

The Figures are mostly good, although I found Figure 1 difficult to read even at 150% magnification. Please check the captions of all figures, as some are not accurate, eg Fig 1. In Figure 2, I suggest the use of a different plot symbol for each of the compounds, ditto in Figure 3.

Apart from the missing references listed above, the Bibliography is extensive and generally well formatted. However there are quite a few errors :-

Charlson et al. 1987 incorrect author name and missing journal

DiTullio et al 2003, missing journal name

Gondwe et al. 2003, errors in the sub and super scripts

Gabric et al 2005 incorrect journal name

Kloster et al 2005, missing journal name

Montes-Hugo et al 2009, problem with volume numbering

Suggest use of ENDNOTE to ensure format consistency.

---

## Author Comment (AC1)

**Answer to Reviewer 1**

*This is an comprehensive report on two austral summer field campaigns that examined the biogenic compound DMS, and its oxidation by-products MSA and nssSO4- at a coastal Antarctic site ( Northern Victoria Land ), in the vicinity of a polynya in the Ross Sea. The atmospheric measurements are augmented by seawater data on DMSP the precursor to DMS, together with various phytoplankton-related biological processes and parameters.*

*The manuscript treats an important topic as data on these compounds is quite limited in the Southern Ocean and Antarctic. The influence of the nearby polynya is also noteworthy.*

**I'd like to thank Prof. Gabric for his positive comments to the paper.**

*One aspect I think that could be improved is the relationship of aerosol concentration with sea ice change. Although the data on sea ice evolution is presented it is not discussed much in relation to the evolution of aerosols at the sampling sites. In Lines 405-409 the authors suggest the high MSA concentrations that preceded the DMS and phytoplankton peaks and may be due to long-range transport. For an alternative view which involves the release of these sulfur compounds during the period of sea ice melt, please see the recent analyses in the SO by Gabric et al. 2005, GBC v19, and in the Arctic by Gabric et al 2018 BAMS, v99.*

**I completely agree with this comment, I just want to say that MSA in this case arise by area of sea ice melting far from the sampling site (not from the very near sea), may be the text was not clear and it is changed in:**

**"At this time MSA and nssSO$_4^{2-}$ were very likely transported from areas far from the sampling site, where an early phytoplankton bloom was taking place likely due to the sea ice melting in the external boundary of sea ice belt around Antarctica (Gabric et al., 2005; Gabric et al., 2018)".**

*The overall presentation is well structured and clear, and except for some minor issues the language is quite good. The document seems to have too many paragraphs which may be a formatting issue which arose in conversion to PDF --not sure.*

*However, the manuscript could be improved by a careful edit, and the authors should check their use of prepositions (English usage is annoyingly different to Italian here), some examples being :-*
*"presents the results on…" should be, "presents the results of…"*
*"DMS is formed in the breakdown…" should read, "DMS is formed by the breakdown.."*
*"the need of long-term measurements.." should read, " the need for long-term measurements"*
*"We separated data in two classes.." should read, "We separated data into two classes"*

**Sorry for this, the manuscript will be revised by a native speaker and only the useful paragraphs are maintained.**

*The Figures are mostly good, although I found Figure 1 difficult to read even at 150% magnification. Please check the captions of all figures, as some are not accurate, eg Fig 1. In Figure 2, I suggest the use of a different plot symbol for each of the compounds, ditto in Figure 3.*

**Figures are redrawn as suggested and captions corrected.**

*Apart from the missing references listed above, the Bibliography is extensive and generally well formatted. However there are quite a few errors :*

*Charlson et al. 1987 incorrect author name and missing journal*
*DiTullio et al 2003, missing journal name*
*Gondwe et al. 2003, errors in the sub and super scripts*
*Gabric et al 2005 incorrect journal name*
*Kloster et al 2005, missing journal name*
*Montes-Hugo et al 2009, problem with volume numbering*
*Suggest use of ENDNOTE to ensure format consistency.*

**You are right, this is due to fault in the use of Mendeley, references will corrected in the new version.**

---

## Author Comment (AC3)

**Answer to Reviewer 2**

*This manuscript describes measurements made over two summers of sulfur containing aerosols, atmospheric DMS mixing ratio, and seawater DMSP concentrations at the Mario Zucchelli Station in Terra Nova Bay, Antarctica. The authors link the aerosol types to biogenic DMS sea-to-air fluxes. The data from the two summers are compared and contrasted. The authors conclude that polynya close to the station is the most important for producing DMS and subsequent fresh aerosols. They also determine a branching ratio of DMS oxidation for freshly produced aerosols to be ~0.84 (older airmasses have a lower ratio). Biogenic particulate matter makes up an average of 17% (max of 56%) of PM10 measured at the station during the summer season. The article contains interesting and useful data, which are sparse, so it should definitely be published. It is generally well written and the techniques/analyses are robust. There are a few minor to moderate issues that should be addressed before publishing. One general comment is that there a number of minor English errors, so the article should be proofread again before submission.*

**Firstly, I'd like to thank Prof. Marandino for the positive comments, the constructive suggestions, and the new references. I think the paper will improve a lot in the final version.**

*Specific comments:*

*Lines 40-41 – DMSO is overlooked by the authors. The cycling of the three compounds (DMS/P/O) is fast and significant (e.g., Archer et al., 2001).*

**A sentence (and related references) on the importance of DMSO is added here in the introduction and in the discussion about the pattern of DMSP in sea water and DMS into the atmosphere.**

**At line 40:**

**"In surface waters, substantial quantities of dissolved DMSP and DMS can be detected, but another important sulfur cycle compound in sea water is dimethylsulfoxide (DMSO) whose concentrations   exceeds the concentration of DMS and DMSP (Asher et al., 2017). DMSO is mainly produced from photochemical and bacterial DMS oxidation, the latter process may serve as an energy source for bacteria (Bodein et al., 2011). The loss processes of dissolved DMS include (i) microbial consumption, (ii) photooxidation, (iii) air-sea gas exchange and 30 (iv) vertical export by mixing (Simo, 2004)."**

**In the discussion of figure 3:**

**… the fast biological turnover (leading to the formation of DMSO) helped in producing low DMS concentrations in summer…**

*Lines 45-47 – Why did the authors pick these reactants and citations? There is a more complete description in the recent publication by Fung et al. (2022) that also includes O3 in the water and Cl in the atmosphere.*

**Thanks for this suggestion, the reference is used here and above. In particular, here the sentence is changed in:**

**"Once in the atmosphere, DMS is oxidized both in gas and water phase by $O_3$ or by the hydroxyl (OH), nitrate ($NO_3$), Chlorine (Cl), and bromine oxide (BrO) radicals to form either methanesulfonic acid (MSA) or sulfur dioxide ($SO_2$), which is further oxidized to $H_2SO_4$ (Gondwe et al., 2003; Read et al., 2008; Fung et al., 2022)."**

*Paragraph starting on line 74 – This is not true. DMS gas exchange has more of a linear relationship with wind speed, as it is largely transported by interfacial exchange and is not as influenced by bubbles (i.e. whitecaps, Bell et al., 2017; Zavarsky et al., 2018) as other more insoluble gases. This is also true for lines 258-259 and 270-271. The discussed decrease in exchange around 10 m/s winds is not universal (Bell et al., 2013; 2015; Yang et al., 2016; Blomquist et al., 2017; Zavarsky et al., 2018). Also, the argument from lines 272-275 (and line 372) is tenuous, especially without DMS measurements, as DMS in the water is more important for the flux than the wind (Marandino et al., 2007).*

**I fully agree that DMS in sea water is the main factor affecting DMS concentration into the atmosphere, but precisely because I do not have the DMS measurements, I felt the need to exclude the wind speed (and therefore the sea-atmosphere transfer speed) among the factors determining the highest concentrations of DMS in the atmosphere in one campaign respect to the other. I reinforced the discussion following the referee suggestions, stressing about the need of DMS measurements in sea water to understand these processes.**

*Line 100 – Should this include other types of aerosol (e.g., primary organics)?*

**Yes, both secondary inorganic and primary organic are important, but in this paper we focus on inorganic secondary aerosol. This is specified in the text as follow:**

**"It is necessary to fill the data gap in the knowledge of biogenically-derived aerosols (both secondary inorganic and primary organics) in the Antarctic to improve understanding of the effects of ocean ecosystem on the marine aerosol-cloud-climate system."**

*Line 115 – Why is the Hulswar et al. (2022) climatology referenced here, but everywhere in the introduction only the earlier Lana (and Kettle) climatology is referenced?*

**I inserted a wrong reference, the correct one is Hulswar et al., 2021.  I use this reference throughout the paper, and I deleted the reference to older climatologies:**

Hulswar, S., Simo, R., Galí, M., Bell, T., Lana, A., Inamdar, S., Halloran, P. R., Manville, G., and Mahajan, A. S.: Third Revision of the Global Surface Seawater Dimethyl Sulfide Climatology (DMS-Rev3), Earth Syst. Sci. Data Discuss., 2021, 1-56, 10.5194/essd-2021-236, 2021.

*Figure 1 – Formatting is different for reporting the years: 2018/2019 vs 2019-2020; typo – the caption says: figure on top reports enlarged map but should say figure on bottom shows enlarged map.*

**Yes, will be correct.**

*Section 2.4 – I think it was a big oversight to not measure DMS in the water. Why was it not done? DMS can be measured reliably after a few hours of storage.*

**You are right, unfortunately we do not have the instrumentation to perform the measurements of the DMS in sea water. These measurements would have been really useful to understand better these processes. In several part of the paper we stress for the need of these measurements to confirm some speculations we do on the basis of the data we have.**

*Lines 239-240 – Did the authors see the recent work on new DMS oxidation products and pathways? Perhaps this factors in to the calculation? The work is also referenced in the Fung et al. (2022) paper.*

**At these lines, the MSA, nssSO$_4^{2-}$ and DMS mean values are presented and we only add the suggested reference. Above in the paper, discussion on the new oxidation products and pathways is added. Thanks for the suggested references that I didn't know.**

*Line 288 – Are there not more recent studies corroborating this? I think this citation can be used, but it is very old. There is a modelling study by Bock et al. (2021) that may also shed some light on this issue.*

**Also considering the above comment, the text is here rearranged and simplified by adding more recent references:**

**"…Therefore, large differences in the DMS sea to air transfer velocity coefficient (k$_w$) do not seem to occur, and are not expected to be the cause of the different behavior of DMS evolution in the two ACs. Unfortunately, we do not have measurements of DMS in sea water, that could have confirmed this hypothesis, but it is consistent with previous modeling and experimental evidences assessing that the large year-to-year variability of atmospheric DMS concentrations can not be explained by changes of meteorological processes controlling the k$_w$ factor or by changes of atmospheric oxidants, but most likely by changes in oceanic DMS concentrations (Sciare et al., 2000; Kettle and Andreae, 2000; Marandino et al., 2007; Bock et al., 2021)."**

*Lines 290-291 – I would tend to agree with this statement, but it really needs a chemical transport model and measurements of DMS in the ocean to corroborate.*

*Lines 292-294 – As the authors mention a bit below, the correlation between biomass and DMS is not reliable (e.g., summer paradox, Simo and Pedros-Alios, 1999). Again, I think not having DMS measurements in the water is a large oversight. In general, these statements are fine (not wrong), but this piece is missing.*

**The sentences at line 290-294 are changed stressing the need of DMS measurements in sea water.**

*Paragraph starting on line 298 (and elsewhere e.g., lines 364-367) – The use/analysis of the back trajectories and the reaction times seems too little (line 375 on) – were trajectories over the whole region check and compared with all data (measured/downloaded)? Please also see my comments to lines 403 on.*

**See answer to the comments below (at line 403)**

*Figure 2 and surrounding text – it is well known that the relationship between DMSP and DMS (in the water) is not straightforward – not only phytoplankton type but also physiology is important (temperature, light levels, etc.). DMS should be measured when possible and DMSO would have been a helpful measurement. Also, it has been observed that there can be a mismatch between DMS water and atmospheric levels, which has to do with physics and chemistry in the atmosphere. Maybe these discussions can be used to point to what future measurements may help elucidate the connections (that were missing here).*

**The sentence is rewritten clarifying the role of processes in sea water, in the conclusions we highlight the importance of future measurements of DMS in sea water to understand these processes.**

Lines 381-382 – Are there physical explanations here (what is influencing this chemistry) that could be mechanistically helpful?

**Here the suggested reference Fung et al., 2022 allow to improve the discussion, the following discussion is added here:**

**In these conditions, the MSA/nssSO$_4^{2-}$ ratio ranges from 0.68 to 0.94 mol/mol, corresponding to percentages of DMS loss up to 41% and 49% respectively for MSA and nssSO$_4^{2-}$ formation. As reported by Fung et al. (2022) the BrO reaction with DMS in gas phase and O$_3$ reaction in aqueous phase are the two main processes for DMS loss in southern high latitude ocean accounting for 50-60% and 20-30 % respectively. Both these processes lead to the formation of MSA in aerosol phase (Fung et al., 2022). In particular, the reaction with O$_3$ in aqueous phase could be particularly efficient at this time when high relative humidity is measured (100%). This high MSA/nssSO$_4^{2-}$ ratio can be measured only in the freshly formed secondary biogenic**

aerosol as MSA in aerosol phase can be transformed in nssSO$_4^{2-}$ by reaction with OH radicals (Fung et al., 2022) leading to a decrease of the MSA/ nssSO$_4^{2-}$ ratio in the aged aerosol.

*Figure 6 – What are the trajectory durations?*

**Trajectories duration is 10 days that will be reported also in the figure caption in the revised version of the paper.**

*Lines 403-420 – Why are the reaction/transport times only considered important sometimes (no. 1) and not at other times (no. 2)? Even with fresh DMS emissions there will be a reaction time and the DMS laden air masses will be transported away before the relevant products form.*

**I agree with this, I think the text was not clear, I insert the following figure reporting the box model I've in mind and these further explanations.**

[Figure]

**Lines 370 and following:**

These processes can be summarized by the simple box model reported in figure, where the box represent the atmosphere over the sampling site (and relative concentration of DMS, MSA and nssSO$_4^{2-}$). F are the flux of DMS, MSA and nssSO$_4^{2-}$ incoming (F…-in), outcoming (F…-out) or formation (F…ox). DMSsw and DMS$_{LR}$ represent the concentration of DMS in sea water and

from sea water far from the sampling site (long range) respectively. $MSA_{LR}$ and $nssSO_4^{2-}{}_{LR}$ represent the concentration of long range transported species and $nssSO_4^{2-}{}_{volc.}$ represent the volcanic $nssSO_4^{2-}$.

From 16 to 17 December 2019 we assumed that $F_{DMS-in}$ is constant and quite high; at the beginning of DMS emission (16 December) $Fox\text{-}MSA$ and $Fox\text{-}nssSO_4^{2-}$ are low therefore the concentration of DMS in the box depend by the equilibrium between $F_{DMS-in}$ and $F_{DMS-out}$. At this wind speed $F_{DMS-out}$ is probably low respect to $F_{DMS-in}$ as the concentration $DMS_{sw}$ (driving the sea-air flux) is high. When $Fox\text{-}MSA$ and $Fox\text{-}nssSO_4^{2-}$ become relevant (at constant $F_{DMS-in}$ and $F_{DMS-out}$) the concentration of MSA and $nssSO_4^{2-}$ increases and DMS decreases, indeed as described above we experimentally find that

DMS emitted = MSA+nssSO4 +DMS residual

Therefore, in this particular situation of constant wind speed and direction the 17 December we can suppose that flux of MSA and $nssSO_4^{2-}$ in and out are negligible and the concentration of MSA and $nssSO_4^{2-}$ in the box are mainly due to the $Fox\text{-}MSA$ and $Fox\text{-}nssSO_4^{2-}$ and for this reason reflecting the $MSA/nssSO_4^{2-}$ ratio of freshly formed biogenic aerosol.

In the following day (18 December) the abrupt change of wind direction (Figure 5) transport on the sampling site different air masses, therefore progressively increasing $F_{DMS-out}$, $F_{MSA-out}$ $F_{nssSO4-out}$ leading to a MSA, $nssSO_4^{2-}$ and DMS concentration decreases in the box of figure.

Lines 204 and followings:

In case 1, due to the presence of sea ice in the proximity of the sampling site $F_{DMS-in}$ is low, therefore is low also the concentration of DMS in the box. MSA and $nssSO_4^{2-}$ come from high $MSA_{LR}$ and $nssSO_4^{2-}{}_{LR}$ therefore from the DMS produced far from the sampling site.

Case 2, in this case due to high variability of wind speed condition and values up to 25-30 m/sec $F_{DMS-in}$ can be high, but in this condition is also high $F_{DMS-out}$ therefore the DMS laden air masses are transported away before a relevant amount of product (MSA and $nssSO_4^{2-}$) are formed. Therefore, even if a small amount of measured MSA and $nssSO_4^{2-}$ in the box can come from $Fox\text{-}MSA$ and $Fox\text{-}nssSO_4^{2-}$, the main part come from $F_{MSA-LR}$ and $F_{nssSO4-LR}$. As $nssSO_4^{2-}{}_{LR}$ can arise from the further oxidation of MSA in water phase along the transport ($F_{MSA-ox}$, Fung et al., 2022) and from volcanic sources ($nssSO4_{volc.}$) the measured $MSA/nssSO_4^{2-}$ ratio is lower and more variable respect to those measured in freshly formed biogenic aerosol.

*Figure 8 – Are the two slopes related to any other metadata – timing, location, air mass origin?*

The split in the two data means low and high biogenic aerosol load and it is related both to the air masses direction and timing. About the 60% of data with nssSO4 concentration lower than 3 $\mu$Mol/m$^3$ comes from the ice sheet (direction from 200-350°N as above reported), the remaining 40% are related to a time before the beginning of sea ice melt and therefore before the phytoplanktonic bloom. This is now reported in the text.

*Line 465 – What is meant by "…reliable…branch ratio."?*

**I mean characteristic for freshly formed biogenic aerosol; the sentence is changed in:**

**"we believe that it is possible to obtain a characteristic MSA/nssSO$_4^{2-}$ branch ratio for the freshly formed biogenic sulfur oxidized aerosol."**

**References**

Archer, S. D., Widdicombe, C. E., Tarran, G. A., Rees, A. P., and Burkill, P. H.: Production and turnover of particulate dimethylsulphoniopropionate during a coccolithophore bloom in the northern North Sea, Aquatic Microbial Ecology, 24, 225-241, https://doi.org/10.3354/ame024225, 2001.

Bell, T. G., De Bruyn, W., Marandino, C. A., Miller, S. D., Law, C. S., Smith, M. J., & Saltzman, E. S. (2015). Dimethylsulfide gastransfer coefficients from algal blooms in the Southern Ocean.Atmospheric Chemistry and Physics,15(4), 1783–1794.https://doi.org/10.5194/acp-15-1783-2015

Bell, T. G., De Bruyn, W., Miller, S. D., Ward, B., Christensen, K. H., & Saltzman, E. S. (2013). Air-sea dimethylsulfide (DMS) gas transferin the North Atlantic: Evidence for limited interfacial gas exchange at high wind speed.Atmospheric Chemistry and Physics,13(21),11,073–11,087. https://doi.org/10.5194/acp-13-11073-2013

Bell, T. G., Landwehr, S., Miller, S. D., de Bruyn, W. J., Callaghan, A. H., Scanlon, B., et al. (2017). Estimation of bubble-mediated air–sea gasexchange from concurrent DMS and CO2transfer velocities at intermediate–high wind speeds.Atmospheric Chemistry and Physics,17(14), 9019–9033. https://doi.org/10.5194/acp-17-9019-2017

Blomquist, B. W., Brumer, S. E., Fairall, C. W., Huebert, B. J., Zappa, C. J., Brooks, I. M., et al. (2017). Wind speed and sea state dependenciesof air-sea gas transfer: Results from the high wind speed gas exchange study (HiWinGS).Journal of Geophysical Research: Oceans,122,8034–8062. https://doi.org/10.1002/2017JC013181

Bock, J., Michou, M., Nabat, P., Abe, M., Mulcahy, J. P., Olivié, D. J. L., Schwinger, J., Suntharalingam, P., Tjiputra, J., van Hulten, M., Watanabe, M., Yool, A., and Séférian, R.:

Evaluation of ocean dimethylsulfide concentration and emission in CMIP6 models, Biogeosciences, 18, 3823–3860, https://doi.org/10.5194/bg-18-3823-2021, 2021.

Fung, K. M., Heald, C. L., Kroll, J. H., Wang, S., Jo, D. S., Gettelman, A., Lu, Z., Liu, X., Zaveri, R. A., Apel, E. C., Blake, D. R., Jimenez, J.-L., Campuzano-Jost, P., Veres, P. R., Bates, T. S., Shilling, J. E., and Zawadowicz, M.: Exploring dimethyl sulfide (DMS) oxidation and implications for global aerosol radiative forcing, Atmos. Chem. Phys., 22, 1549–1573, https://doi.org/10.5194/acp-22-1549-2022, 2022.

Marandino, C. A., De Bruyn, W. J., Miller, S. D., & Saltzman, E. S. (2007). Eddy correlation measurements of the air/sea flux of dimethylsulfideover the North Pacific Ocean.Journal of Geophysical Research,112, D03301. https://doi.org/10.1029/2006JD007293

Yang, M., Bell, T. G., Blomquist, B. W., Fairall, C. W., Brooks, I. M., & Nightingale, P. D. (2016). Air-sea transfer of gas phase controlledcompounds.IOP Conference Series: Earth and Environmental Science,35(1), 012011.

Zavarsky, A., Goddijn-Murphy, L.,Steinhoff, T., & Marandino, C. A. (2018).Bubble-mediated gas transferand gas transfer suppressionof DMS and CO2.Journal ofGeophysical Research: Atmospheres,123, 6624–6647.https://doi.org/10.1029/2017JD028071

**Citation**: https://doi.org/10.5194/acp-2022-195-RC2

**Most of these references will be quoted and discussed.**

---

## Author Response (AR1)

UNIVERSITÀ
DEGLI STUDI
FIRENZE

**UGO SCHIFF**
DIPARTIMENTO
DI CHIMICA

Dear Editor,

I'm sending you the revised version of the paper: "**Factors controlling atmospheric DMS and its oxidation products (MSA and nssSO$_4^{2-}$) in the aerosol at Terra Nova Bay, Antarctica**" by Silvia Becagli et al.

Referees' suggestions are all appreciated and accepted, I thank both the referees for their support to this paper.

The point-to-point answer to the referee's comments and the related changes made to the manuscript are reported in the specific answers to each referee, here only the answers to the main comments are reported.

The main criticisms are expressed by referee 2 and in summary are: (i) the missing data of DMS in sea water and (ii) a deep discussion on the fate of atmospheric DMS to better explain the trend of measured concentration of DMS, MSA and nssSO4 in the atmosphere at the sampling site.

Regarding the first point, I completely agree, but unfortunately we don't have such measurements, therefore in several parts of the paper (including in the abstract and in the conclusion) I stress about the need for DMS measurements in sea water to better understand the DMS concentration trend into the atmosphere. Besides, according to the referee suggestions I improve the discussion on the sea-to-atmosphere DMS transfer as function of wind speed also referring to the published paper suggested by the referee.

"Besides processes in the water column, ocean-air DMS flux has a more of a linear relationship with wind speed, as it is largely transported by interfacial exchange and it is not as influenced by bubbles (i.e. whitecaps, Bell et al., 2017; Zavarsky et al., 2018) as other more insoluble gases. Vlahos and Monahan (2009) evidenced that at wind speed higher than 10 m/sec DMS transfer rates decreases due to the amphiphilic nature of DMS that leads to transfer delay because higher wind speeds cause a higher concentration of sinking bubbles by whitecapping of the ocean surface. Anyway, Marandino et al. (2007) demonstrate that most of the variance in the fluxes can be accounted by variations in DMS sea surface concentration (37%) than wind speed (19%).

.....

Therefore, large differences in the DMS sea to air transfer velocity do not seem to occur and are not expected to be the cause of the different behavior of DMS evolution in the two ACs. Unfortunately, we do not have measurements of DMS in sea water, that could have confirmed this hypothesis, but it is consistent with previous modeling and experimental evidences assessing that the large year-to-year

variability of atmospheric DMS concentrations can not be explained by changes of meteorological processes controlling the $k_w$ factor or by changes of atmospheric oxidants, but most likely by changes in oceanic DMS concentrations (Sciare et al., 2000; Kettle and Andreae, 2000; Marandino et al., 2007; Bock et al., 2021)."

To answer to second point a figure reporting the scheme of the processes leading to the measured concentration of DMS, MSA and nssSO4 is added and discussion is enlarged referring to this figure and other published results suggested by the referee.

"These processes can be summarized by the simple model reported in Fig. 5, where the box represent the atmosphere over the sampling site (and relative concentration of DMS, MSA and $nssSO_4^{2-}$). $F_{DMS}$, $F_{MSA}$, and $F_{nssSO4}$ are the flux of DMS, MSA and $nssSO_4^{2-}$ incoming (F-in), outcoming (F-out) or formation (Fox). $DMS_{sw}$ and $DMS_{LR}$ represent the concentration of DMS in sea water and from sea water far from the sampling site (long range) respectively. $MSA_{LR}$ and $nssSO_4^{2-}{}_{LR}$ represent the concentration of long range transported species and $nssSO_4^{2-}{}_{volc.}$ represent the volcanic $nssSO_4^{2-}$.

[Figure]

**Figure 5. Schematic representation of the processes related to the measured concentration of DMS, MSA and nssSO4$^{2-}$ at MZS. See text for the abbreviations meaning.**
……
……
…..

Looking at the scheme in Fig. 5, from 16 to 17 December 2019 we assumed that $F_{DMS-in}$ is constant and quite high; at the beginning of DMS emission (16 December 2019) $F_{ox-MSA}$ and $F_{ox-nssSO4}$ just started therefore the concentration of DMS in the box depend by the equilibrium between $F_{DMS-in}$ and $F_{DMS-out}$.

At this wind speed $F_{DMS-out}$ is probably low respect to $F_{DMS-in}$ as the concentration $DMS_{sw}$ (driving the sea-air flux) is high. In these conditions, the DMS concentration reached a maximum of 32.8 nMol/m$^3$ on 16 December (average over the period 9:00-21:00 LT). Due to the constant wind speed and direction (Fig. 6) we can assuming that DMS emission from the ocean remains constant also in the following days ($DMS_{emitted}$), when UV radiation increase in the following 24 h stimulated the DMS oxidation processes, $F_{ox-MSA}$ and $F_{ox-nssSO4}$ become relevant and at constant $F_{DMS-in}$ and $F_{DMS-out}$, the concentration of MSA and $nssSO_4^{2-}$ increases and DMS decreases. MSA and $nssSO_4^{2-}$ reached the maximum concentration on December 17 (8.3 and 9.9 nMol/m$^3$ for MSA and $nssSO_4^{2-}$, respectively, for the sampling time 9:00-21:00 LT), when the DMS concentration was 12.6 nMol/m$^3$.

Therefore, the December 17 in the box of Fig. 5 we should have:

$DMS_{emitted} = DMS_{lost} + DMS$

If DMS emitted = 32.8 nMol/m$^3$ and DMS in the box is 12.6 nMol/m$^3$

$DMS_{lost}$ = 32.8 - 12.6 = 20.2 nMol/m$^3$

But the $DMS_{lost}$ is due to the formation of MSA and $nssSO_4^{2-}$ therefore:

$DMS_{lost}$ = MSA+$nssSO_4^{2-}$ = 8.3 + 9.9 = 18.2 nMol/m$^3$ that is in agree with the value of 20.2 previously calculated with the approximation of the constant DMS emission in these days.

Therefore, in this situation of constant wind speed and direction the 17 December we can suppose that $F_{MSA}$ and $F_{nssSO42-}$ in and out are negligible and the concentration of MSA and $nssSO_4^{2-}$ in the box are mainly due to the Fox-MSA and Fox-$nssSO_4^{2-}$, for this reason reflecting the MSA/$nssSO_4^{2-}$ ratio of freshly formed biogenic aerosol.

In the following day (18 December) the abrupt change of wind direction (Figure 5) transport on the sampling site different air masses, therefore progressively increasing $F_{DMS-out}$, $F_{MSA-out}$ $F_{nssSO4-out}$ leading to a MSA, nssSO42- and DMS concentration decreases in the box of Fig. 5."

Referee 1 suggests to give more importance to the sea ice melting in area far from the sampling site to explain early concentration peaks of biogenic aerosol ("…At this time MSA and $nssSO_4^{2-}$ were very likely transported from areas far from the sampling site (i.e. $MSA_{LR}$ and $nssSO_4^{2-}{}_{LR}$ -Fig.5) where an early phytoplankton bloom was taking place likely due to the sea ice melting in the external boundary of sea ice belt around Antarctica (Gabric et al., 2005; Gabric et al., 2018)." ) and to modify the figure in order to easily read them. This is accomplished by re-drawing the pictures with different and enlarged symbol for each parameter (now not only differing by colour) when necessary.

Here an example of the new aspect of figure 6.

[Figure]

[Figure]

Finally, the paper was revised for English language and checked for references.
I hope the paper is now acceptable for publication on ACP.

Best regards,
Silvia and Co-Authors